# Electrical Cortical Stimulation for Language Mapping in Epilepsy Surgery—A Systematic Review

**DOI:** 10.3390/brainsci15121267

**Published:** 2025-11-26

**Authors:** Honglin Zhu, Efthymia Korona, Sepehr Shirani, Fatemeh Samadian, Gonzalo Alarcon, Antonio Valentin, Ioannis Stavropoulos

**Affiliations:** 1Department of Basic and Clinical Neuroscience, Institute of Psychiatry, Psychology and Neuroscience, King’s College London, London SE5 8AB, UK; 2Department of Clinical Neurophysiology, King’s College Hospital NHS Foundation Trust, London SE5 9RS, UK; 3Max Planck UCL Centre for Computational Psychiatry and Ageing Research, Queen Square Institute of Neurology, University College London, London WC1E 6BT, UK; 4Department of Clinical Neurophysiology, Royal Manchester Children’s Hospital, Manchester M13 9WL, UK

**Keywords:** epilepsy surgery, electrical cortical stimulation, language mapping, language tasks, cortical language areas

## Abstract

**Background:** Language mapping is a critical component of epilepsy surgery, as postoperative language deficits can significantly impact patients’ quality of life. Electrical stimulation mapping has emerged as a valuable tool for identifying eloquent areas of the brain and minimising post-surgical language deficits. However, recent studies have shown that language deficits can occur despite language mapping, potentially due to variability in stimulation techniques and language task selection. The validity of specific linguistic tasks for mapping different cortical regions remain inadequately characterised. **Objective:** To systematically evaluate the validity of linguistic tasks used during electrical cortical stimulation (ECS) for language mapping in epilepsy surgery, analyse task-specific responses across cortical regions, and assess current evidence supporting optimal task selection for different brain areas. **Methods:** Following PRISMA [2020] guidelines, a systematic literature search was conducted in PubMed and Scopus covering articles published from January 2013 to November 2025. Studies on language testing with electrical cortical stimulation in epilepsy surgery cases were screened. Two reviewers independently screened 956 articles, with 45 meeting the inclusion criteria. Data extraction included language tasks, stimulation modalities (ECS, SEEG, ECoG, DECS), cortical regions and language error types. **Results:** Heterogeneity in language testing techniques across various centres was identified. Visual naming deficits were primarily associated with stimulation of the posterior and basal temporal regions, fusiform gyrus, and parahippocampal gyrus. Auditory naming elicited impairments in the posterior superior and middle temporal gyri, angular gyrus, and fusiform gyrus. Spontaneous speech errors varied, with phonemic dysphasic errors linked to the inferior frontal and supramarginal gyri, and semantic errors arising from superior temporal and perisylvian parietal regions. **Conclusions:** Task-specific language mapping reveals distinct cortical specialisations, with systematic patterns emerging across studies. However, marked variability in testing protocols and inadequate standardisation limit reproducibility and cross-centre comparisons. Overall, refining and standardising the language task implementation process could lead to improved outcomes, ultimately minimising resection-related language impairment. Future research should validate task–region associations through prospective multicentre studies with long-term outcome assessment.

## 1. Introduction

Surgical resection of the epileptogenic zone is an effective treatment for a selected group of patients with focal drug-resistant epilepsy [1]. Nonetheless, this procedure can give rise to post-surgical impairment in crucial cognitive functions, particularly language, due to resection in proximity to eloquent cortical areas or disruption of white matter connectivity. Temporal lobectomy or corticectomy and resection of lesions of the frontal or parietal lobe can result in language impairment or a further decline in pre-existing deficit.

Recent advances in brain anatomy have led to the understanding that cortical functions such as language are generated by a network of dispersed specialised cortical regions that are interconnected by white matter fibres which are arranged anatomically in bundles [2]. Areas of the cortex that are involved in the integration of the language function are the frontal lobe (frontal operculum and precentral gyrus), temporal lobe (inferior, middle, and superior gyri), and parietal lobe (postcentral gyrus and supramarginal gyrus) [3,4]. Moreover, the motor component of speech is associated with sites concentrated in the pre- and postcentral gyri [5]. Language deficits occur when cortical areas or white matter connections are damaged. There is heterogeneity in the structure of these networks that is more diverse among the population with epilepsy due to potential functional reorganisation in response to brain pathology [2,6,7]. Therefore, functional mapping is crucial for identifying critical language areas, ultimately preserving communication abilities after surgery, enhancing neuropsychological outcomes and post-surgical quality of life [8,9,10].

There is a lateralisation of cortical regions associated with language over one hemisphere, most commonly the left. Traditionally, language lateralisation is assessed pre-surgically by the intracarotid sodium amobarbital test (Wada test), now largely replaced by fMRI [11]. Extraoperative electric cortical stimulation (ECS), which utilises subdural electrodes, is the gold standard for functional mapping [11,12,13]. ECS during stereoelectroencephalography (SEEG), a modality of intracranial EEG recording using depth electrodes and targeting both superficial and deep cortical regions, can serve as an alternative for language mapping preoperatively. Additionally, intraoperative language mapping by direct cortical stimulation (DECS) and electrocorticography (ECoG) enables evaluation of the language function by stimulation of the cortex and subsequent testing of the patient with language tasks as well as recording for possible induction of epileptiform discharges (after-discharges) or ictal activity [14].

The language function is tested by stimulating possible language areas and interpreting possible deficits after stimulation, in the absence of after-discharges. Examples of language tasks used during electrical stimulation include visual confrontation naming, verbal/semantic fluency, auditory description naming, token test, spontaneous speech, word repetition, object description, reading, syntax, auditory comprehension, counting, and verb generation [15].

Various language deficits, such as speech arrest, comprehension deficits, paraphasia, agraphia, alexia, anomia, aphasia, apraxia of speech, and tone disruption, may be elicited during stimulation, reflecting disturbances of function in different eloquent areas or connections. Eliciting the aforementioned deficits requires a cautious assessment of various elements of language function. This can be warranted only if a careful selection of specific language tasks is implemented, followed by a succinct interpretation of their outcomes.

Despite routine performance of language mapping in clinical practice, the absence of a standardised protocol leads to discrepancies in electric stimulation methods, choice of language tests, surgical resection margins, and definition of critical language regions across centres [16]. Restricted operation time dictated by infection risk and patient tolerance constrains the number of tests that can be conducted during intraoperative monitoring. Post-surgical language deficits can appear despite functional mapping, potentially due to inadequate and inappropriate task selections or mistakes in interpretation [16,17].

Several studies have investigated cognitive tests in neurosurgery. In a comprehensive review, Hamberger et al. (2007) established the foundations which have guided clinical practice for epilepsy surgery for nearly two decades [18]. However, the field has developed significantly since 2007, especially with the global widespread application of SEEG techniques [8,9,19]. In 2013, De Witte et al. conducted a systematic review of intraoperative language testing [20], but their primary focus was on brain tumour surgery. However, when comparing with epilepsy surgery, the challenges are fundamentally different because of different patterns of cortical reorganisation [21]. Ruis conducted a comprehensive review of intraoperative cognitive tests during awake craniotomy for epilepsy and glioma, though the selection of language tests was not evaluated [22]. Lu et al. explored mapping strategies for counting and picture-naming practices, but primarily restricted to these two tasks [23]. More recently, Reecher et al. reviewed language mapping of electrical stimulation language in paediatric epilepsy [9]. However, there is no recent review exploring the use of SEEG in adult and paediatric epilepsy surgery during the last decade.

This review aims to systematically evaluate the validity of linguistic tasks used during electrical cortical stimulation (ECS) for language mapping in epilepsy surgery, analyse task-specific responses across cortical regions, and assess current evidence supporting optimal task selection for different brain areas.

## 2. Methods

### 2.1. Protocol and Registration

The systematic literature search was conducted in PubMed and Scopus from January 2013 to October 2025. We followed the PRISMA 2020 guidelines when conducting and writing up this review.

The review is registered with Open Science Framework (OSF, 10.17605/OSF.IO/2M3UT).

### 2.2. Eligibility Criteria

Studies were considered eligible according to the following PICO framework:

Population (P): Adults or children with epilepsy undergoing epilepsy surgery evaluation with electrical cortical stimulation for language mapping.

Intervention (I): Language testing during electrical cortical stimulation, including extraoperative ECS with subdural electrodes, stereoelectroencephalography (SEEG), intraoperative direct electrical cortical stimulation (DECS), or electrocorticography (ECoG).

Comparator (C): Not applicable (descriptive review of task–region associations).

Outcomes (O): Primary outcomes included the identification of cortical regions producing specific language deficits during stimulation and the correlation between linguistic tasks and anatomical regions. Secondary outcomes included stimulation parameters and error types.

### 2.3. Search Strategy

The systematic literature review searches for a combination of “epilepsy” AND “mapping” AND “electric stimulation”. The exact search terms used are (epilepsy [Title/Abstract]) AND (brain mapping [MeSH Terms]) OR (language [Title/Abstract]) OR (language mapping [Title/Abstract]) OR (language test*[Title/Abstract]) OR (speech [Title/Abstract]) OR (cognitive test*[Title/Abstract]) OR (cognitive mapping [Title/Abstract])) AND (neurosurgical procedures [MeSH Terms]) OR (electric stimulation [MeSH Terms])). Two reviewers (HZ and EK or FS and SS) independently screened titles/abstracts and assessed full texts against predefined criteria. Disagreements were resolved by consensus or, if needed, by a third reviewer (AV or IS). Data was extracted by HZ, EK, and FS from the included reports and entered into a structured Excel database.

### 2.4. Study Selection and Data Extraction

A total of 956 papers in English were identified. Manual screening was carried out by title and abstract (n = 78), then by full text (n = 45), adhering to the following inclusion criteria (Figure 1):

Eligible studies reported task-induced language effects (e.g., speech arrest, anomia, comprehension, or repetition errors) associated with a specific cortical site. We included only English-language, primary observational studies/cohorts published between 2013 and 2025. Epilepsy had to be the predominant aetiology; therefore, we included studies where epilepsy was reported among other causes only where seizure type-specific data for epilepsy were available. Papers exclusively addressing other pathologies, like tumours, irrespective of their potential association with seizures, were excluded.

A manuscript was rejected if a report named language disturbances without the related description of which task (e.g., “speech arrest” or “reading difficulty”) was affected by an error. Studies using solely non-invasive modalities (fMRI or TMS) were excluded; when multiple modalities were used, only ECS-, ECoG-, or SEEG-derived data were extracted.

In the synthesis, we grouped studies by task and cortical region, and a study contributed to a grouping only when it reported both the stimulated site and a task-specific language effect. Anatomical labels and units were harmonised to a common nomenclature; missing or unclear fields were coded as “not reported,” with no data imputation and no author contact. Due to heterogeneity in tasks, stimulation parameters, and outcome definitions, we did not conduct a meta-analysis and instead used a structured narrative synthesis. Heterogeneity was explored qualitatively by modality (ECS versus SEEG), operative setting (intraoperative versus extraoperative), age group, hemisphere, language dominance, and reporting quality. No formal sensitivity analyses were undertaken because no quantitative pooling was performed. Formal statistical tests for reporting bias were not feasible; potential publication and selection biases are discussed qualitatively in the limitations section.

## 3. Results

Among the 45 included papers, a total of 1196 participants were studied, including at least 381 adults and 390 paediatric participants (aged under 18). Many of the papers did not clarify the age of the patients included. A total of 837 patients had subdural electrodes, 328 had SEEG, and 31 had both subdural electrodes and SEEG. A wide array of language tasks are employed across centres. While spontaneous speech, visual naming, and auditory naming are the most frequently used tasks, none of them are present in every paper. Auditory comprehension, repetition, reading, the token test, and verb generation are some other tests used across centres (Table 1).

Visual confrontation naming has been the cornerstone of language testing during electrical stimulation. Its process comprises the stages of identification, access to semantic representation, and lexicalisation. In our review, visual naming deficits primarily manifested in the following areas: middle temporal gyrus (MTG), inferior temporal gyrus (ITG), and posterior temporal cortex (PTC), posterior frontal cortex (PFC), fusiform gyrus (FG), parahippocampal gyrus (PHG), and occasionally in postcentral gyrus (PoG), Heschl’s gyrus HG, and insula [16,28,29,30,31,32,34,35,36,37,38,39,56,61]. Auditory naming assesses phonological processing and semantic associations and is also widely performed. Naming impairment was elicited around posterior frontal cortex (PFC), posterior temporal cortex (PTC), especially the posterior superior temporal gyrus (pSTG) and posterior middle temporal gyrus (pMTG), and occasionally also angular and fusiform gyrus [15,37,55,60] (see Figure 2 for anatomical regions).

Different tasks focusing on examining the semantics aspect of language function were applied. These would implement either completion of a sentence regarding the category of an object, or “who” and “what” questions about specific categories of such as objects, animals etc, or the identification of semantic characteristics of objects shown to patients. They were associated with deficits in the posterior superior temporal gyrus (pSTG), middle temporal gyrus (MTG), and angular gyrus (AnG) [16,37,45]. Auditory comprehension is based on semantic processing and speech perception. Two of the studies implemented verbal commands, whereas in another, token test was performed. Areas that showed dysfunction during stimulation were the inferior temporal gyrus (ITG) and the fusiform gyrus (FG) [39,41,46,62].

Spontaneous conversation was occasionally tested and most of the time used as a supplementary assessment along with visual naming. It was commonly used as an assessment task in paediatric population, and a simple recitation of the alphabet or numbers was also used in many studies as a screening task to test electrode sites. Errors were only occasionally segregated into different categories. A broad spectrum of areas was identified as relevant. From the studies that could specify the type of dysphasic errors, errors in expressive speech were more commonly associated with the posterior inferior frontal gyrus, inferior precentral gyrus, supramarginal gyrus, and angular gyrus, whereas receptive speech errors were stimulated from the superior temporal gyrus and adjacent perisylvian parietal cortex [29,36,49,50].

Verb generation was only embodied in two studies, even though it is known to have a wider cortical anatomical substrate than object generation. Studies’ results review showed the implication of multiple cortical areas, such as inferior frontal gyrus, precentral gyrus, postcentral gyrus, angular and supramarginal gyrus, as well as the posterior temporal cortex and right temporal lobe [25,36]. Areas associated with verb generation overlapped to an extent with areas implicated in the reading task.

Reading was assessed in a great proportion of the studies. It requires multiple language functions, both lexical–semantic processing and phonological generation, but identification of errors, which could further associate a specific dysfunction with a cortical area, was not described in many of them. Areas involved in producing deficits included the posterior inferior frontal gyrus (pIFG), parahippocampal gyrus (PHG), fusiform gyrus (FG), superior temporal gyrus (STG), and inferior temporal gyrus (ITG) [16,27,34,39,40,42,43,45,47,48,58,59]. Writing testing was rarely performed, and it was associated with the lateral fusiform gyrus and posterior temporal cortex stimulation [44,49]. Repetition of words or syllables was examined in three studies and affected by the stimulation of the inferior frontal gyrus (IFG) and the posterior superior temporal gyrus (pSTG) [47,51,60].

Phonemic production and tone were not routinely tested as isolated features. Two studies which examined tone found deficits in lateral inferior frontal gyrus and posterior middle frontal gyrus [49,50]. A specific phonological task found association with posterior and middle temporal gyrus as well as supramarginal gyrus [37]. Phonological errors were also elicited by stimulation of pars opercularis (opIFG), middle frontal gyrus, hippocampus, and inferior and middle temporal gyri [58]. Interestingly, isolated phonemic and semantic fluency tasks were not implemented during electrical stimulation in any of the studies in the current review.

Most studies identified language-related areas in the left hemisphere. A few studies identified bilateral language-related areas in left-handed patients. More specifically, activation of the right basal temporal and right superior temporal gyrus was identified with reading in a multilingual left-handed right-dominant hemisphere patient [61]. Bilateral activation of the supramarginal gyrus could result in dysarthria or speech arrest [30,66].

Beyond associations between language tasks and stimulated regions, the current review focused on the identification of errors during lingual tasks between studies. We identified that in some studies, dysarthria, an orofacial sensorimotor impairment caused by muscle contractions in the tongue or throat, was classified as a language impairment [17,29,46], whereas in others as a false positive language response due to its intrinsic sensorimotor nature [50,67]. It is worth noting that cortical stimulation may identify the negative motor area (NMA), which causes cessation of voluntary movement and can be misinterpreted as a speech arrest. Distinguishing the NMA, typically located inferior to the precentral gyrus, from frontal language areas is crucial [68]. Moreover, there was convergence but not consensus between studies in defining criteria for language areas. Most papers designate language areas as sites where disruption is repeatedly induced [15,28,39,42,49,50,53,55,57,62,67,69,70], whereas one stated that disruption must occur in over 75% of trials for a site to be considered positive [38], while another follows a centre-specific protocol [36].

Regarding the population of the participants, most studies included only adult patients, some included only paediatric patients [17,35], and some had a mixed population [30,31,32]. As mentioned above, spontaneous speech and visual naming were the language tasks preferentially used in paediatric population. Cross-linguistic validation demonstrates consistent cortical localisation patterns across English, French (DO80 picture-naming), and Japanese (Kanji and Kana reading) protocols, supporting universal principles of cortical language organisation [28,54].

To summarise the results of this systematic review, impairments in visual confrontation naming are likely to be elicited after stimulation of left inferior temporal gyrus, inferior frontal gyrus, parahippocampal gyrus, and fusiform gyrus [26,60]. Auditory naming was associated with deficits in the left posterior superior temporal gyrus and posterior middle temporal gyrus, as well as angular and fusiform gyrus. Errors in spontaneous speech were associated with a wide network of frontal and parietal regions, such as the left inferior frontal gyrus, inferior precentral gyrus, supramarginal gyrus, and angular gyrus, especially regarding phonemic dysphasic errors, and left superior temporal gyrus and adjacent parietal cortex regarding semantic errors. Identification of errors in a systematic way was not performed in a proportion of studies. Reading was commonly assessed and identified a wide range of areas of both temporal, frontal, and parietal regions, most commonly inferior frontal, posterior temporal regions, and supramarginal gyrus (Figure 3).

Details on test modalities used in each included paper can be found in Table 2.

## 4. Discussion

### 4.1. Methodological Heterogeneity and Clinical Implications

There are several different aspects of language functions. Application of linguistic tasks to language examination traditionally included expressive and receptive speech, repetition, writing, and reading. The traditional anatomical basis of speech, in which expressive speech is integrated in Broca’s area in the left frontal area, and receptive speech in Wernicke’s area in the left posterior temporal area, has already been challenged. Language cortical stimulation studies have mostly used three main functional regions: the anterior language area, posterior language area, and basal temporal language area, which was first described in patients with epilepsy by Luders et al. [26,72,73,74]. Most recently, fMRI studies have used a model of six core clinically relevant areas: Broca’s area, Exner’s graphemic motor area (posterior middle frontal gyrus), the supplementary speech area within dominant SMA/pre-SMA, the angular gyrus, the posterior superior temporal/temporoparietal ‘Wernicke’ region, and the basal temporal language area (fusiform/inferior temporal) [75].

Recent data support the idea of functionally specific language modules integrated into networks, with significant cortical-to-cortical and cortical-to-subcortical connections. In this context, even simple tasks such as the retrieval of a single word are subserved in distributed cortical areas [58,76]. The linguistic tasks implemented during mapping can be analysed depending on the language functions required for their accomplishment: visual decoding, semantic processing, lexical access, generation of phonemes, and articulatory sequencing [17].

In this review, we focused on language tasks used during electric stimulation. Although data were not extracted for other testing modalities such as fMRI or transcranial magnetic stimulation (TMS), several studies have compared those modalities against electric stimulation. Hirano et al. and Austermuehle et al. tested the concordance of results between fMRI and ECS [42,62]. Language fMRI is an established method for the lateralisation of language, but lacks accuracy in identifying eloquent areas. Even though recommendations exist among centres for the language tasks implemented during fMRI testing, a considerable range of variability in their quantity and content was also noted. Word generation tasks followed by comprehension tasks were more widely used but were also found to have strengths and limitations, the description of which would be beyond the scope of this paper [77]. The use of common paradigms during multimodal language mapping could further validate the association between specific linguistic tasks and their anatomical substrates. 

One significant step to resolve this heterogeneity is the creation of a standardised language test tailored for short periods of brain stimulation. For instance, Ohlerth et al. showed that both object and action-naming tasks can be completed within nTMS/DES temporal constraints (≥50 items per task with ≥80% naming agreement) by validating a comprehensive battery (VAN-POP) in 80 healthy adults across three languages. Before being widely used, these protocols must be validated in clinical populations with epilepsy and correlated with intraoperative and postoperative findings [64].

The studies included in this review utilised either extraoperative or intraoperative ECS and/or electrocorticography for language mapping. During intraoperative language mapping, naming tasks are effective but may lead to prolonged operation times [35,45,78]. Conversely, spontaneous speech tasks, such as counting, dialogue, or reciting, allow for continuous action with minimal pauses and are executed quickly. This approach is more natural, potentially reducing patient stress while enhancing cooperation and tolerance. The reviewed results suggest that spontaneous speech can effectively assess the posterior inferior frontal gyrus, parietal lobe, and posterior superior temporal gyrus. This aligns with previous research indicating that spontaneous speech during electrocorticography is a sensitive task for frontal regions, alongside with visual naming [32,50]. The strength of using spontaneous speech lies in its ability to capture more language errors compared to specific tests isolating different functions. On the other hand, automatic speech assessment, such as counting during ECS, lacks sensitivity in assessing basal temporal areas, as the cessation of counting was identified in only 21% of the posterior regions [79].

Besides task selection, the choice of electrode type also determines mapping accuracy. Electrical cortical stimulation (ECS) using subdural electrodes is the gold standard with high specificity but has limited spatial sampling and omits cortices within sulci, leading to underestimated maps [47,51]. Stereo-EEG (sEEG) provides access to sulcal and deep cortical regions, enabling both stimulation mapping and task-related recordings. It can extend language mapping to regions inaccessible by subdural grids and improve coverage of distributed networks [30,80]. However, potential limitations include relatively sparse sampling along each electrode trajectory, dependent on a pre-implantation hypothesis. These constraints may reduce sensitivity if critical contacts are missed. Comparative evidence demonstrates that ECS with subdural electrodes yield the highest spatial resolution for defining essential language areas, while SEEG has a complementary role for sulcal and deep network, especially in a paediatric setting or in lesional cases with complex topography, as suggested by Cossu et al. [81].

One particular problem of functional mapping is that the risk of after-discharges and seizure induction can also hamper feasibility, particularly near seizure foci. To avoid this problem, high-gamma electrocorticography (ECoG) is a technique that can analyse localised task-related power increases in the high-gamma range, and can delineate eloquent cortex without stimulation, avoiding the risk of after-discharges [47,53]. It offers broader spatial coverage than ECS and high temporal resolution; however, its sensitivity and specificity vary across studies, and it is currently recommended as a supplementary rather than a standalone technique [29,42].

### 4.2. Stimulation Parameters and Protocols

The protocols for stimulation techniques applied through various centres were found to be inconsistent. Bipolar pair-electrode stimulation was typically utilised, with a stimulation pulse width ranging from 100 to 500 microseconds and a 50 Hz frequency, rarely reaching 60 Hz. Stimulation intensity began at 1 mA and could reach a maximum of 17.5 mA with a stepwise increment of 0.5–2 mA. It was observed that the majority of language deficits were induced when testing at the highest current amplitudes. Additionally, a correlation was noted between the number of errors and the amplitude of stimulation, highlighting the significance of amplitude in identifying areas functionally involved in language production. Possible underlying mechanisms may involve the impact on white matter tracts or the disruption of cortical networks [42]. These findings underscore the importance of carefully considering stimulation parameters in ECS language studies to better understand the functional implications on language production.

Heterogeneity was also noticed in defining critical language errors. While complete speech arrest is deemed a critical impairment, there are disagreements on the significance of signs like hesitation and perseveration [16]. Some studies categorised impairments as complete (e.g., speech arrest) or partial (e.g., paraphasia), influencing subsequent management strategies [49,50]. Moreover, the threshold for the number of errors required to label sites as critical also varies, ranging from 25% to 100% [16]. Additionally, not all studies applied segregation of paraphasic errors as semantic or phonemic, as proposed by Corina et al. [82]. Finally, only one centre could offer a formal assessment of the participant’s responses by neuropsychologist [60]. Undoubtedly, a type of error-based language mapping would contribute to the identification of associations between lingual functions and cortical areas.

### 4.3. Surgical Margins and Resection Planning

The choice of operational margins is also crucial. There was a concordance among centres and a 1 cm distance from the functional margin is generally accepted for resection [38,83]. The same distance is supplemented on glioma resection surgeries near functional pathways after electrical stimulation [14]. This highlights the importance of carefully balancing the need for complete resection with the need to preserve important functional areas.

From a surgical perspective, converging data from ECS, SEEG, and ECoG should guide resections beyond predetermined surgical margins, highlighting functional boundaries derived from customised, language-appropriate tasks [84,85]. Functional atlases for language mapping can be further improved by combining intraoperative and intracranial (sEEG or subdural) test results.

### 4.4. Age-Specific Considerations: Paediatric Versus Adult Language Mapping

Special considerations are needed when mapping is applied to different populations. Diversity in developmental and cognitive presurgical status, age of epilepsy onset, as well as limitations in terms of cooperation and tolerability of the exam, makes language mapping in paediatric population challenging [17]. There is no concordance in the degree of overlap of cortical areas in language mapping between the adult and paediatric populations [86,87]. The reasons why paediatric results frequently differ from adult data can be attributed to cortex development during the epileptic process, which can reflect on a neuroplastic reorganisation of fronto-temporal networks [88]. To prevent underestimating or overestimating eloquent regions in younger patients, age-stratified interpretation and stimulation threshold adjustment could be essential [89].

### 4.5. Cross-Linguistic Considerations

A multimodal study of language localisation and integration of a left-handed multilingual patient showed activation of the right superior temporal gyrus for at least the three languages examined [30]. The results of this analysis showed that even though the conventional standard-of-care for language mapping is based on the use of cued response-inhibition paradigms, the use of additional language tasks is strongly suggested and associated with better post-surgical neuropsychological outcomes [29,31].

Additionally, caution is warranted for patients whose native language is not English, as different language systems may require different skillsets hence different tasks. For instance, tone of speech is of particular importance in Chinese language [49,50], which is generated in the laryngeal motor cortex [90]. In Japan, Kanji and Kana, language systems utilising different skills, are tested separately. Details are extensively outlined in studies by Enatsu, Hirano, and Shimotake [27,40,62]. Cross-linguistic research demonstrates that English-based task paradigms might not transfer to other linguistic systems, as tonal and logographic languages use partially different cortical circuits, such as increased activation of laryngeal motor and occipitotemporal regions [91,92]. Reducing cultural bias and increasing the accuracy of functional localisation are two benefits of adapting mapping tasks to native language structure.

### 4.6. Developmental Language Reorganisation

Importantly, early onset dominant temporal lobe epilepsy is associated with more widespread and atypical distribution of language areas and the recruitment of more anterior temporal regions [93]. The transfer of language functions can occur not only through the same hemisphere but also interhemispherically in chronic epilepsy. This was also evident in one study in our review where significant language reorganisation was identified from the dominant to the non-dominant hemisphere [61]. Lesional and non-lesional epilepsy can result in different language organisational patterns [6].

Language tests in epilepsy and other neurological conditions are often studied collectively [22]. Future studies should explore variations in language tasks required for assessment of patients with different pathologies, taking into consideration variations that might occur due to different aetiology and cortical excitability [50]. Previous comparative analyses on optimal extraoperative language tasks in tumours and epilepsy were limited in the range of paradigms analysed [94]. Studies evaluating tasks beyond the commonly studied spontaneous speech and naming tasks would be valuable.

### 4.7. Strengths and Limitations of This Review

The strength of this review lies in its comprehensive systematic review, offering insights into language task selections for each brain region based on the structural–functional relationship, thus providing guidance to maximise testing efficiency. However, a limitation is the potential incompleteness of recorded language tasks in the studies included, especially when their primary focus is not on language testing [58]. Papers that specifically investigated functions of eloquent areas often employed numerous tasks [37,40,44]. Neither of the above would accurately reflect routine clinical practice. Moreover, the variability in language tasks, the stimulation parameters, the interpretation of the findings, and the use of subdural or depth electrodes make the grouping and analysis of the data difficult and reflect further limitations of this work. More specifically, the stimulation parameters play a significant role in the ECS findings, especially in the paediatric population, and this might have caused a bias when grouping findings from study cohorts of different ages. A recent review of findings of electrical stimulation language mapping in children describes these difficulties in detail [9]. Finally, the evidence base is likely affected by publication and selective reporting bias, with positive or striking effects more likely to be published and variability in outcomes reducing the visibility of null or inconclusive findings.

## 5. Conclusions

We present a systematic review of the specific language task selection and their association with different cortical areas, discussing key factors in mapping, including testing modality, definition of language error, identification of critical language area, and considerations regarding different populations. Despite the application of advanced neurophysiological techniques in language mapping, an individualised approach remains a challenge due to variability in the localisation of language sites, which is more pronounced in the epilepsy population and the lack of a standardised protocol for assessing the constellation of lingual elements that are essential for speech production. By integrating evidence-based task selection and a multimodal approach, the accuracy of language mapping can be optimised along with our understanding of language network organisation. A unified battery of expressive and receptive tasks, combining visual and auditory naming, reading or counting, and repetition, while always considering the patient’s multilingual background and age, should be used. Moreover, multimodal assessment combining invasive and non-invasive tools could increase accuracy and prevent possible mistakes inherent to specific techniques. Overall, refining and standardising the language task implementation process could lead to improved outcomes, ultimately minimising resection-related language impairment. Further multicentre studies using comparable language testing protocols are required to provide insight into the organisation of language networks and provide accurate mapping and better post-surgical outcomes.

## Figures and Tables

**Figure 1 brainsci-15-01267-f001:**
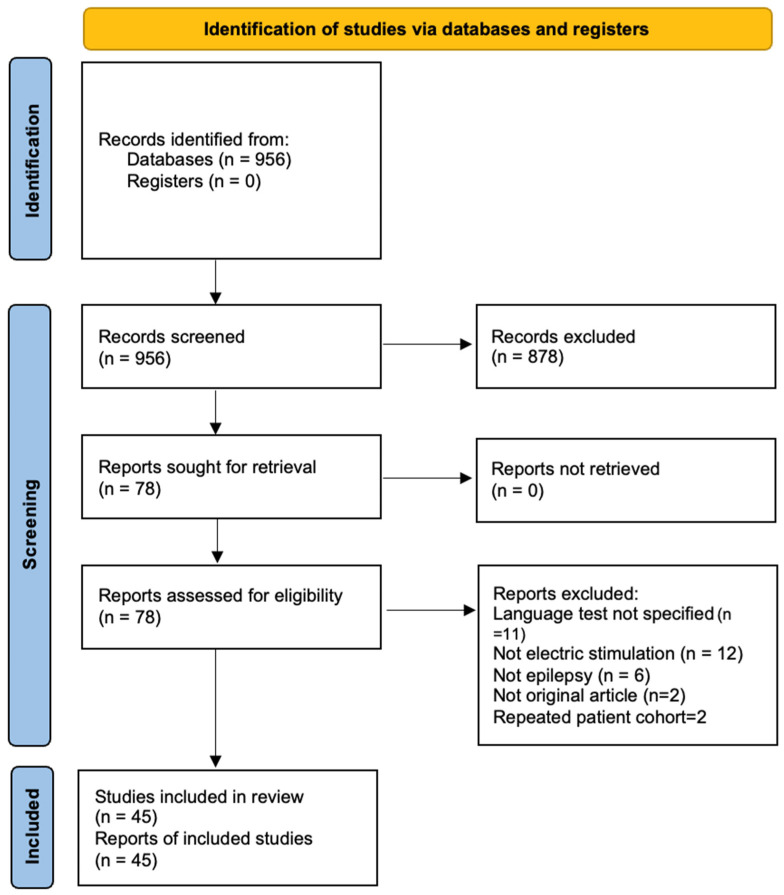
PRISMA flow chart illustrates the identification and selection process [24].

**Figure 2 brainsci-15-01267-f002:**
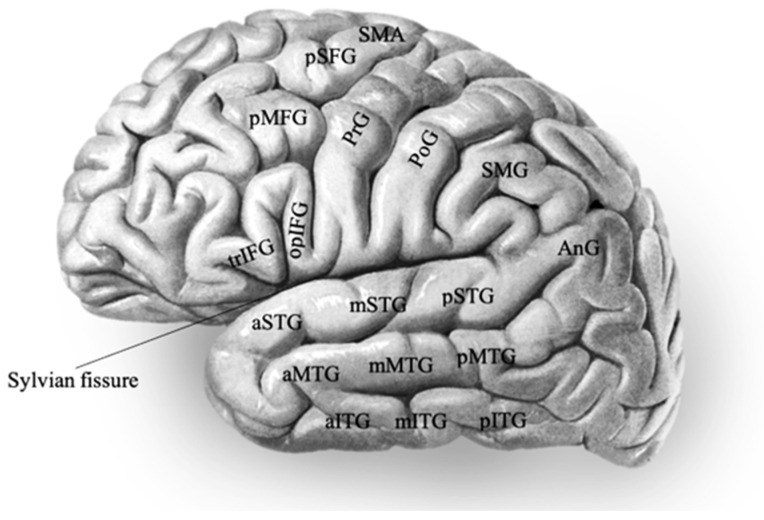
Diagram on language-related brain regions with abbreviations. Original picture adapted from radiopedia.org. Frontal lobe: IFG—Inferior frontal gyrus, pIFG—Posterior inferior frontal gyrus, trIFG—Pars triangularis, opIFG—Pars opercularis, pMFG—Posterior middle frontal gyrus, pSFG—Posterior superior frontal gyrus, PFC—Posterior frontal cortex (including pIFG, pMFG, pSFG), PrG—Precentral gyrus, SMA—Supplementary motor area; temporal lobe: STG—Superior temporal gyrus, MTG—Middle temporal gyrus, ITG—Inferior temporal gyrus, a/m/pSTG—Anterior/middle/posterior superior temporal gyrus, a/m/pMTG—Anterior/middle/posterior middle temporal gyrus, a/m/pITG—Anterior/middle/posterior inferior temporal gyrus, HG—Heschl’s gyrus (not visible in the diagram), PTC—Posterior temporal cortex (including pSTG, pMTG, pITG), PHG—Parahippocampal gyrus (not visible in the diagram), FG—Fusiform gyrus (not visible in the diagram); parietal lobe: PoG—Postcentral gyrus, SMG—Supramarginal gyrus, AnG—Angular gyrus. Adapted from illustration from “Sobotta’s Textbook and Atlas of Human Anatomy” 1907 [65], now in the public domain.

**Figure 3 brainsci-15-01267-f003:**
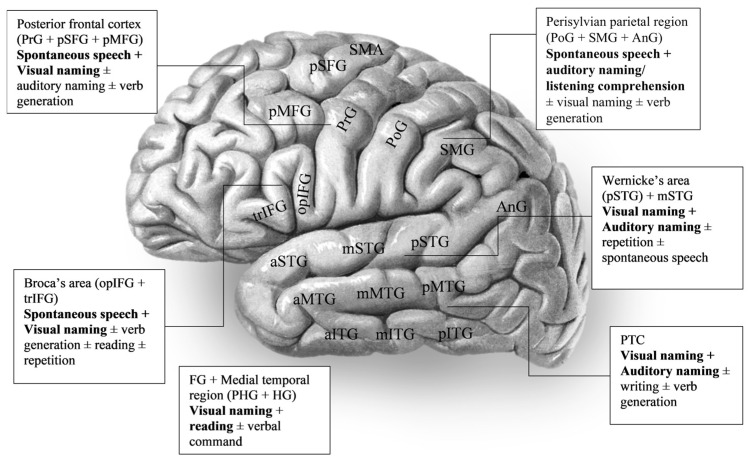
Language map on task selection for brain cortices adapted from radiopedia.org. Where ± is used it indicates the test is not essential. Frontal lobe: trIFG—Pars triangularis, opIFG—Pars opercularis, pMFG—Posterior middle frontal gyrus, pSFG—Posterior superior frontal gyrus, PrG—Precentral gyrus, SMA—supplementary motor area; temporal lobe: a/m/p/STG—anterior/mid/posterior Superior temporal gyrus, a/m/pMTG—anterior/mid/posterior Middle temporal gyrus, a/m/pITG—anterior/mid/posterior Inferior temporal gyrus; HG—Heschl’s gyrus (not visible in the diagram), PTC—Posterior temporal cortex (including pSTG, pMTG, pITG), PHG—Parahippocampal gyrus (not visible in the diagram), FG—Fusiform gyrus (not visible in the diagram); parietal lobe: PoG—Postcentral gyrus, SMG—Supramarginal gyrus, AnG—Angular gyrus. Adapted from illustration from “Sobotta’s Textbook and Atlas of Human Anatomy” 1907 [65], now in the public domain.

**Table 1 brainsci-15-01267-t001:** Description of language paradigms and error definition.

Linguistic Task	Description of the Test and Time of Delivering the Test	Error Appreciation/Comments	Literature Reference
**Visual object naming**	Patients were shown pictures of common items (e.g., umbrella) and instructed to say, “This is a…,” Pictures were either line drawings or coloured photographs In most studies, patients were trained to name each picture (control trials) to ensure task feasibility and picture familiarity (baseline level). One study involved semantic odd picture naming where patient identified one picture out of four that did not fit and explained why ([25]) Examples in other languages include the French visual naming task (DO80) [23,26], visual naming task for Kanji and Kana nouns [27] Covert (silent) naming was asked in one centre examining paediatric patients	Content of errors In most studies, anomia or paraphasia was considered abnormal. In two studies, errors were subcategorised as semantic, phonological, or mixed. In some, dysarthria (sensorimotor component) was considered abnormal. Speed reduction was considered abnormal in two studies and hypophonia in one. Memory (amnestic anomia) declines were considered in one study. More types of errors noted in one study: apraxia, neologisms, perseverations. Quantity of errors Sometimes an error in more than one trial per task would be assumed abnormal. If results were ambiguous, additional trials were administered. In some, “reproducible changes” were considered abnormal.	[9,11,12,18,23,24,25,26,27,28,29,30,31,32,33,34,35,36,37,38,39,40,41,42,43,44,45,46,47,48,49,50,51,52,53,54]
**Auditory description naming**	Descriptive cue for the target word delivered aloud. The cue is read and named by the patient within the stimulation period. Descriptions of objects using <10 words: e.g., “a household pet that purrs” Stimulation occurs immediately after the item is presented, and before the oral response of the carrier phrase by the patient [49] Used both intra- and extraoperatively	Inaccuracy. Sites were deemed critical if the patient failed to name items during stimulation but gave correct responses upon cessation of stimulation.	[11,35,37,38,46,47,49,52]
**Auditory comprehension**	Examples of questions asked: “Do flamingos stand on one leg?” Or assessed by asking the patient to follow one or two-step verbal commands, e.g., finger tipping, hand or tongue movement, simple arithmetic calculations	Yes/No answer. Unable to follow commands.	[36,37,42,43,44,46,50,51]
**Comprehension (auditory + semantics)**	Pt was asked to answer brief auditory questions: e.g., ‘What flies in the sky? All questions, beginning with either ‘what’, ‘where’, ‘when’ or ‘who’, were designed to elicit one- or two-word answers with nouns	Patients were instructed to answer, “I don’t know” when they did not know the answer to or did not understand a question. In case they failed to verbalise a relevant answer when asked for the reason. Neuropsychologists present to assess.	[55]
**Single-word auditory comprehension (SWAC) test**	Patients are asked to listen to a target word and describe what it means without saying the target word Stimulation is delivered as the patient listens to the target word	Error if the patient mentioned that the target word was not understood, a word was not spoken or asked if the target word could be repeated.	[15]
**Token test**	Patients pointed at objects given the experimenter’s verbal description Ex: “Point to the small, red circle”	Right/wrong.	[40,41,56]
**Semantic orientated testing**	Patients were presented with pictured objects (1 per stimulation trial) and instructed to indicate whether objects had one of the following semantic features: e.g., “found indoors”, “something people typically eat”, “a musical instrument”, “found in a garden”	Yes/No answer.	[34]
**Sentence completion**	“A bow is used to...” “You blow air into...” Stimulation occurs immediately after the item was presented, and before the oral response of the carrier phrase by the patient [49] Used intraoperatively		[25,49]
**Phonological test**	Patients were instructed to indicate whether the pictured objects begin with a particular sound (first phoneme): e.g., “Does this begin with the sound “t” as in toy?” Used extraoperatively	Yes/No answer.	[11,34]
**Spontaneous conversation**	Three conditions were identified: speaking, listening, and rest Having a dialogue on an everyday topic [57] Duration was 3 min Used intraoperatively	Speech arrest. Detection of slight changes in language performance (decreased fluency and inarticulacy) Not described thoroughly in most studies.	[23,25,28,36,42,48,51,58]
**Numerical and/or Alphabet recitation**	It is identified as part of spontaneous conversation in one study Word list recall in one study [56] Sequence naming of months of the year in one study [40] Reciting poetry in two studies on Mandarin language testing [57,58] Used intraoperatively	A reproducible functional change, such as a pause or dysarthria.	[36,47,48,55,56,58,59,60]
**Reading**	Patients were instructed to read short sentences aloud In studies on the Japanese language, morphograms (Kanji) reading, and syllabograms (Kana) reading tasks were employed as well [27,40]	Speech arrest [59]. In one study [46], error types were categorised as follows: slow/effortful reading (apraxia), syntactic errors, sentence stem additions, sentence stem omissions, mixed sentence stem errors (additions + omissions).	[30,31,36,37,40,41,42,43,44,45,46,48,49,50,51,56,60]
**Verb generation**	Auditory presentation of a series of nouns and instructed to covertly generate as many associated verbs as possible	Arrest or interruption of speech.	[28,61]
**Repetition**	Sentence, word or consonant–vowel syllable repetition (e.g., /da/)	Speech arrest.	[25,39,44,46,47,51,55,58,62]
**Story processing**		Yes/No answer.	[61]
**Sensorimotor** **Component testing**	Humming		[55]
**Automatic speech recognition**	Passive listening to a corpus of short sentences (Texas Instruments/Massachusetts Institute of Technology (TIMIT)		[63]
**Written word–picture matching**	Japanese version adapted from the Cambridge 64-item semantic battery Performed extraoperatively		[40]
**Text question response task**	Overtly provide answers to questions presented in text Performed extraoperatively		[25]
**Writing**	Used a validated Chinese version of the Western Aphasia Battery (C-WAB)		[49,50]
**Tone production monitoring**	Tone of speech in Mandarin was monitored as patients were undertaking other tests, e.g., spontaneous speech or picture naming		[50]
**Animal sound recognition and naming**	Patients are asked to listen to typical sounds produced by animals and name them, e.g., barking sound produced by a dog Performed extraoperatively		[47]
**Verb and noun test for peri-operative testing (VAN-POP)**	Black-and-white drawings of person/animal performing action with lead-in phrase. Patient completes with verbs inflected for person, number, tense. Used for nTMS and DES	Anomias, paraphasias, speech arrest, grammatical errors (inflection failures), light verb paraphrasing.	[64]
**Spoken word–picture matching**	Six pictures shown. Patients indicate target after hearing spoken name. Intraoperative DES.	Arrest, slowing, incorrect matching. Quantity of errors: Highest impairment rate.	[54]
**Spoken verbal command**	Make gestures following simple spoken instruction. Auditory only. Intraoperative DES.	Slowing, unable to follow commands. Not impaired in PHG.	[54]

**Table 2 brainsci-15-01267-t002:** Studies included with stimulation modalities, language tests used, and task-specific language response sites if applicable.

**Studies**	**Stimulation Modalities**	**Language Tests**	**Task-Specific Language Response Sites**
**Alarcón et al., 2019** [15]	eoECS + ioECS	Single-word auditory comprehension (SAWC) test	L-PTC especially pSTG, AnG
**Alonso et al., 2016 [59]**	SEEG	Reading	Dominant IFG
**Aron et al., 2022 [28]**	SEEG	French visual naming (DO80)	Ventral temporal cortex, including PHG, FG, ITG
**Aron et al., 2024 [19]**	SEEG	French visual naming (DO80)	Anterior/posterior parahippocampal gyrus (PHG), fusiform gyrus (FG), and inferior temporal gyrus (ITG)
**Arya et al., 2015 [29]**	eoECS + eoECoG	ECS—visual naming ECoG—spontaneous speech (dialogue)	Picture naming—L/R-pSTG, L/R inferior PoG, L-pIFG Spontaneous speech (listening)—L-pSTG, L-mSTG, adjacent perisylvian parietal cortex Spontaneous (speaking)—L-inferior PrG, L-pIFG, AnG, SMG
**Arya et al., 2017 [32]**	eoECS + eoECoG	ECS—visual naming ECoG—covert visual naming	L-pMFG, L-pIFG
**Arya et al., 2019 [30]**	SEEG	Picture naming	Picture naming—L/R-STG, L-MTG, L-HG, L-IFG Dysarthria—STG, HG, planum temporale
**Austermuehle et al., 2017 [42]**	eoECS	Counting/alphabet recitation, visual naming, reading, token test	Reading—temporal receptive regions
**Babajani-Feremi et al., 2018 [34]**	eoECS + eoECoG	ECS—sentence reading, comprehension, token task ECoG—overt object naming	L-IFG, STG, MTG, dorsal premotor region, inferior-Rolandic region
**Bauer et al., 2013 [36]**	eoECS + eoECoG	ECS—object naming ECoG—spontaneous conversation, verb generation, picture naming	Spontaneous speech (listening)—inferior Sylvian fissure Spontaneous speech (speaking)—superior Sylvian fissure Verb generation—PTC, AnG, SMG, PrG, PoG, IFG Naming—perisylvian area
**Bearden et al., 2023 [56]**	SEEG	Confrontation naming, word list recall, word list encoding	L-trIFG, hippocampus, anterior temporal pole
**Bohm et al., 2020 [61]**	eoECS	Reading, comprehension, naming in three languages	Naming—PTC
**Cockle et al., 2025 [52]**	SEEG	Visual and auditory naming, reading, spontaneous speech, and counting	Fusiform gyrus, ITG, MTG, STG, temporal pole, entorhinal cortex, pre-SMA
**Enatsu et al., 2017 [27]**	ECS	Reading (paragraph and words)	Paragraph reading—L-FG, L-PHG, L- ITG
**Ervin et al., 2020 [53]**	SEEG	Visual naming (Snodgrass picture set)	Posterior temporal and temporoparietal cortices (bilateral); posterior quadrant HGM sites, often ESM linked to declines in working memory, naming, and verbal learning; no ESM+ sites resected
**Hamberger et al., 2014 [16]**	eoECS	Language task in different centres: speech production (76%), comprehension (68%), naming (89%), reading (75%)	Anterolateral temporal region, basal temporal region, insular cortex
**Hamberger et al., 2016 [37]**	eoECS	Visual naming, auditory naming, semantic task (yes/no to auditory questions regarding pictures), phonological task (yes/no to auditory questions regarding first phoneme of object names)	Phonological tasks—L-pSTG, L-MTG, SMG Semantic tasks—L-pMTG, L-pITG
**Hamberger et al., 2019 [38]**	eoECS + ioECS	Visual naming, auditory naming	Visual naming—mSTG (younger patients), STG and MTG (older patients) Auditory naming—STG (younger patients), STG, MTG, and SMG (older patients)
**Hirano et al., 2020 [62]**	eoECS	Reading sentences, spontaneous speech (counting/ speaking), object naming, auditory comprehension (following verbal commands)	L-pO/pT, p STS, p MTG, and SMG
**Korostenskaja et al., 2014 [25]**	eoECS + eoECoG	ECS—picture naming ECoG—story processing, picture naming (overt/ covert), verb generation	ECS (picture naming)—no area found ECoG—story processing and picture naming in right lateral and basal temporal regions; verb generation in L-frontal and R-temporal lobes
**Labudda et al., 2017 [70]**	eoECS	Object naming, reading, following verbal commands	L IFG + PTG small area in R MFG
**Lee et al., 2020 [55]**	eoECS	Visual naming, auditory naming	Expressive aphasia (auditory naming)—pIFG, PrG, PTC Expressive aphasia (visual naming)—mTG, ITG Receptive aphasia (auditory + naming)—pSTG, pMTG
**Lioumis et al., 2023 [71]**	eoECS	Naming, sentence repetition	Naming—trIFG, opIFG Sentence repetition—pSTG
**Matoba et al., 2024 [54]**	ECS	Picture naming, spoken word–picture matching, Kanji word reading, paragraph reading, spoken verbal command, Kana word reading	Anterior FG and ITG (visual + auditory) semantic impairments; middle FG mainly unimodal (visual) processing; PHG least impaired
**Nakai et al., 2017 [60]**	eoECS + eoECoG	Auditory naming, syllable repetition, humming, counting, reciting alphabet	Auditory naming (receptive aphasia)—L-pSTG, L-pMTG Auditory naming (expressive aphasia)—L-PTC, L-FG Auditory naming (speech arrest)—bilateral inferior PrG, L-pSFG
**Oane et al., 2020 [43]**	SEEG	Reading, counting	Cingulate cortex
**Perrone-Bertolotti et al., 2020 [58]**	ECS + SEEG	Picture naming, reading, etc.	Picture naming (speech arrest)—left frontal region (PrG, opIFG, trIFG, SMA, insula), left temporal region (TG, FG), PoG Picture naming (speech paraphasia)—L-insula, L-TG Picture naming (phonological paraphasia)—opIFG, MFG, TG
**Rolinski et al., 2019 [39]**	eoECS	Continuous recitation task, visual naming, reading, token test	Basal temporal cortex
**Rolston et al., 2018 [3]**	eoECoG	Sentence listening, consonant-vowel syllables repetition (e.g., /da/)	Frontal operculum, precentral gyrus, ITG, MTG, STG, postcentral gyrus, SMG
**Sabsevitz et al., 2020 [44]**	SEEG	Picture naming, auditory naming, famous face naming, syllable repetition, single word reading, writing to dictation	Impaired reading without agraphia—lateral L-FG
**Serafini et al., 2013 [45]**	eoECS	Visual naming, auditory naming, reading, sentence completion	Auditory naming—STG, AnG Visual naming—mMTG, SMG, AnG Sentence completion—STG, MTG, AnG
**Shimotake et al., 2015 [40]**	eoECS + eoECoG	ECS—reading, picture naming, spoken verbal command, spoken and written word–picture matching ECoG—picture naming	All tasks in ITG, anterior FG
**Takahashi et al., 2022 [46]**	eoECS	Sentence reading, spontaneous speech, object naming, verbal command	STG, ITG, SMG, IFG
**Wen et al., 2017 [47]**	eoECoG	Picture naming, animal sound recognition and naming, text question response, auditory question response, word reading, word repetition	Language production—opIFG, pMFG, inferior precentral and postcentral gyrus, SMG Language comprehension—PTC
**Zhou et al., 2021 [69]**	ioECS + ioECoG	ECS—spontaneous speech, comprehension task ECoG—counting, reciting traditional Chinese poems, having dialogues	STG and IFG
**Young et al., 2018 [57]**	SEEG + eoECoG	Verbal fluency (counting, reciting months), visual naming, auditory naming, repetition	ECoG—IFG, STG, MTG, ITG SEEG—STG, pMTG, MFG
**Yu et al., 2018 [48]**	SEEG	Reading/counting	L-PSG (posterior short gyrus) close to the precentral operculum and L-precentral operculum at pIFG
**Zhang et al., 2013 [49]**	eoECS + ioECS	io + eoECS—spontaneous speech, comprehension, visual naming eoECS only—repetition, reading, writing	Writing—L-PTC Repetition—L-IFG, L-pSTG Tone—L-IFG Reading—L-STG, L-pIFG Visual naming—L-PTC
**Zhang et al., 2020 [50]**	eoECS	Picture naming, spontaneous speech, listening comprehension, tone monitor	Tone—R-pMFG in left-handed patient Naming—L-pSTG, L-vPrG, L-pMTG Comprehension—L-STG

Abbreviations: eoECS—Extraoperative electric cortical stimulation; ioECS—Intraoperative electric cortical stimulation; SEEG—Stereoelectroencephalography; L-PTC—Left posterior temporal cortex; m/pSTG—middle/posterior superior temporal gyrus; IFG—Inferior frontal gyrus; PHG—Parahippocampal gyrus; FG—Fusiform gyrus; ITG—Inferior temporal gyrus; PoG—Postcentral gyrus; PrG—Precentral gyrus; AnG—Angular gyrus; SMG—Supramarginal gyrus; MFG—Middle frontal gyrus; MTG—Middle temporal gyrus; HG—Heschl’s gyrus; pO/pT—Posterior occipital/Posterior temporal; STS—Superior temporal sulcus; PTG—Posterior temporal gyrus; trIFG—Pars triangularis; opIFG—Pars opercularis; SMA—Supplementary motor area.

## Data Availability

This article is a systematic review based on data extracted from previously published studies. No new data were generated or analysed in this study. All data used are publicly available in the original publications cited within the article.

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
