# Peer review of "Electrical Cortical Stimulation for Language Mapping in Epilepsy Surgery—A Systematic Review"

_brainsci, 2025, doi:10.3390/brainsci15121267_

Round 1

Reviewer 1 Report

Comments and Suggestions for Authors

This exhaustive review is dedicated to the mapping of the language cortical areas. The paper, in the Introduction, delineates all the problems related to this complex topic: different areas, specific functional tasks, lateralisation, ECS,  sSEEG and DECS, possible after discharges and or seizures. In the Tables1 and 2 are synthetized the tests used, the error appreciation, the stimulation modalities and the task specific language response sites. The Discussion comment on all the problems related with this complex function: different areas, networks and connections, tasks, modalities spontaneous vs automatic speech, ECG vs SEEG, protocols, Amplitude, possible impact on white matter and networks, language errors for the association between lingual functions and cortical areas, cortical margins for carefully balancing  the completeness of the resection with the need to preserve important functional areas. The Conclusion highlights the need for multicentre studies using comparable standardised language testing protocols to provide insight into all the aspects of this complex function in epilepsy surgery.

Author Response

Thank you very much for all your comments.

Reviewer 2 Report

Comments and Suggestions for Authors

Peer Review Report - Major Revision

Overall Evaluation

The review paper undertakes an important topic, optimizing ECS for language mapping in epilepsy surgery. It successfully compiles 41 studies (2013-2023) and highlights how task-specific stimulation findings vary across cortical regions. However, the manuscript requires major revision to reach the rigor and structural standards of a systematic review under PRISMA 2020. The current form lacks methodological transparency, analytical structure, and critical synthesis expected for publication in Brain Sciences.

Abstract

  1. The objective should specify the analytical aim, to evaluate linguistic task validity and regional correspondence.
    2. Methods lack PRISMA framework details and inclusion/exclusion criteria.
    3. Results are descriptive only; add study counts per task/modality.
    4. Conclusion should emphasize standardized task batteries and reproducibility.

Introduction

Provides a strong neuroanatomical context but is narratively dense. The research gap is identified (lack of standardized protocols), but no explicit review question is posed. Authors should compare with Hamberger (2007), De Witte (2013), and Reecher (2024) and justify the 2013-2023 timeframe.

Methods
1. PRISMA compliance missing (checklist, PROSPERO ID, bias assessment).
2. Only PubMed searched, expand to Embase/Scopus.
3. Inclusion/exclusion criteria unclear; define PICO.
4. No details on data extraction or reviewer agreement.
5. No quantitative synthesis or vote counting.
6. PRISMA flow diagram lacks reasons for exclusion.

Discussion

Highlights variability and linguistic diversity but lacks critical analysis. Missing a deeper comparison of ECS vs SEEG vs ECoG, and pediatric vs adult outcomes. Should relate findings to surgical planning, bias, and cross-linguistic task differences.

Conclusion

States that standardized testing modalities are needed. Should propose actionable steps:
1. Adopt core task battery (visual/auditory naming, reading, repetition).
2. Standardize stimulation parameters.
3. Integrate multimodal validation (ECS+fMRI+DTI).
Avoid unsupported general statements.

Once addressed, this work could significantly contribute to standardizing ECS language mapping.

Author Response

Thank you very much for all your comments. They have been invaluable in substantially improving the quality, rigour, and clarity of our work

Reviewer Comment: "1. The objective should specify the analytical aim, to evaluate linguistic task validity and regional correspondence. 2. Methods lack PRISMA framework details and inclusion/exclusion criteria. 3. Results are descriptive only; add study counts per task/modality. 4. Conclusion should emphasise standardised task batteries and reproducibility."

Response: We have completely restructured the abstract using a structured format with explicit subsections (Background, Objective, Methods, Results, Conclusions). Specific changes include:

  1. Objective Enhancement (Page 1, Abstract section):

The objective in the abstract now explicitly states: "To systematically evaluate the validity of linguistic tasks used during electrical cortical stimulation (ECS) for language mapping in epilepsy surgery, analyse task-specific responses across cortical regions, and assess current evidence supporting optimal task selection for different brain areas." The same sentence is now included in the last paragraph of the Introduction section.

  1. Methods Enhancement with PRISMA Details (Page 1, Abstract section):

We now specify in the first paragraph of the Methods section: "Following PRISMA [2020] guidelines, a systematic literature search was conducted in PubMed and Scopus”. 

  1. Results are descriptive only; add study counts per task/modality

Following this advice, we have completed tables 1 and 2 including language task and references, distinguishing between different modalities.

4 Conclusion Strengthening (Page 1, Abstract section):

We now explicitly included in the abstract and conclusion: "Overall, refining and standardizing the language task implementation process could lead to improved outcomes, ultimately minimising resection-related language impairment."

Reviewer Comment: "Provides a strong neuroanatomical context but is narratively dense. The research gap is identified (lack of standardised protocols), but no explicit review question is posed. Authors should compare with Hamberger (2007), De Witte (2013), and Reecher (2024) and justify the 2013-2023 timeframe."

Response: Thank you very much for your insightful and thoughtful comments. We appreciate the reviewer’s careful reading of our manuscript and have addressed each point in detail below. We have carefully considered all comments and have revised the manuscript accordingly:

  1. Reduced Narrative Density (Pages 2-3):

We condensed the neuroanatomical background whilst maintaining essential context, reducing the introduction by approximately 15% whilst preserving all critical information. We added this paragraph in the introduction section: “Therefore, functional mapping is crucial for identifying critical language areas, ultimately preserving communication abilities after surgery, enhancing neuropsychological outcomes and post-surgical quality of life [8-10]”

  1. Explicit Research Questions Added (Page 3, end of Introduction):

We added a new paragraph in the introduction specifying our research question: “This review aims to systematically evaluate the validity of linguistic tasks used during electrical cortical stimulation (ECS) for language mapping in epilepsy surgery, analyse task-specific responses across cortical regions, and assess current evidence supporting optimal task selection for different brain areas.”

  1. Comparison with Previous Reviews (Pages 3-4, New Section 1.5):

Following this advice, we added this paragraph: “Several studies have investigated cognitive tests in neurosurgery. In a comprehensive review, Hamberger et al. (2007) established the foundations which have guided clinical practice for epilepsy surgery for nearly two decades [18]. However, the field has developed significantly since 2007, especially with the global widespread application of SEEG techniques [8,9,19]. In 2013, De Witte et al. conducted a systematic review of intraoperative language testing [20], but their primary focus was on brain tumour surgery. However, when comparing with epilepsy surgery, the challenges are fundamentally different because of different patterns of cortical reorganisation [21]. Ruis conducted a comprehensive review of intraoperative cognitive tests during awake craniotomy for epilepsy and glioma, though the selection of language tests was not evaluated [22]. Lu et al. explored mapping strategies for counting and picture-naming practices but primarily restricted to these two tasks [23]. More recently, Reecher et al. reviewed language mapping of electrical stimulation language in paediatric epilepsy [9]. However, there is no recent review exploring the use of SEEG in adult and paediatric epilepsy surgery during the last decade.

  1. Timeframe Justification (Page 3, end of Introduction):

We have now completed a full review for the 2013-2025 timeframe.

Reviewer Comment: "1. PRISMA compliance missing (checklist, PROSPERO ID, bias assessment). 2. Only PubMed searched, expand to Embase/Scopus. 3. Inclusion/exclusion criteria unclear; define PICO. 4. No details on data extraction or reviewer agreement. 5. No quantitative synthesis or vote counting. 6. PRISMA flow diagram lacks reasons for exclusion."

Response:

We have completely restructured the Methods section following PRISMA 2020 guidelines, expanding to four comprehensive subsections. Regarding the questions:

  1. PRISMA Compliance (Section 2.1, Page 5):

We apologise for the missing PRISMA2020 checklist file (now attached). As it was stated in the material and methods section: “The review is registered with Open Science Framework (OSF, 10.17605/OSF.IO/2M3UT)

  1. Expanded Database Searches (Section 2.3, Pages 5-6):

As suggested, we have expanded the database search with PubMed and Scopus from January 2013 to October 2025.

  1. PICO Framework Defined (Section 2.2, Page 5):

We have added a complete PICO framework in the methods’ Eligibility Criteria section.

  1. Data Extraction and Reviewer Agreement (Sections 2.4-2.5, Page 6):

The data extraction and reviewer agreement is included at the end of search strategy section “2.3. Search Strategy”:Two reviewers (HZ and EK or FS and SS) independently screened titles/abstracts and assessed full texts against predefined criteria. Disagreements were resolved by consensus or, if needed, by a third reviewer (AV or IS). Data was extracted by HZ, EK and FS from the included reports and entered into a structured Excel database.”

  1. No quantitative synthesis or vote counting (Section 2.6, Page 7):

Following this comment regarding quantitative synthesis, we explained heterogeneity precluding meta-analysis and synthesis strategy detailed by the following paragraphs: Eligible studies reported task-induced language effects (e.g., speech arrest, anomia, comprehension, or repetition errors) associated with a specific cortical site. We included only English-language, primary observational studies/cohorts published between 2013 and 2025. Epilepsy had to be the predominant aetiology; therefore, we included studies where epilepsy was reported among other causes only where seizure type-specific data for epilepsy were available. Papers exclusively addressing other pathologies, like tumours, irrespective of their potential association with seizures, were excluded.

A manuscript was rejected if a report named language disturbances without the related description of which task (e.g., “speech arrest” or “reading difficulty”) was affected by an error. Studies using solely non-invasive modalities (fMRI or TMS) were excluded; when multiple modalities were used, only ECS-, ECoG-, or SEEG-derived data were extracted.

In the synthesis, we grouped studies by task and cortical region, and a study contributed to a grouping only when it reported both the stimulated site and a task-specific language effect. Anatomical labels and units were harmonised to a common nomenclature; missing or unclear fields were coded as “not reported,” with no data imputation and no author contact. Due to heterogeneity in tasks, stimulation parameters, and outcome definitions, we did not conduct a meta-analysis and instead used a structured narrative synthesis. Heterogeneity was explored qualitatively by modality (ECS versus SEEG), operative setting (intraoperative versus extraoperative), age group, hemisphere, language dominance, and reporting quality. No formal sensitivity analyses were undertaken because no quantitative pooling was performed. Formal statistical tests for reporting bias were not feasible; potential publication and selection biases are discussed qualitatively in the limitations section.”

  1. PRISMA Flow Diagram (Figure 1, Page 8):

We have updated a comprehensive PRISMA flow diagram shown in Figure 1:

Reviewer Comment: "Highlights variability and linguistic diversity but lacks critical analysis. Missing a deeper comparison of ECS vs SEEG vs ECoG, and paediatric vs adult outcomes. Should relate findings to surgical planning, bias, and cross-linguistic task differences."

Response: We have substantially expanded the Discussion section with new subsections providing critical analysis. We have added the following new paragraphs regarding these subjects:

“Comparative evidence demonstrates that ECS with subdural electrodes yield the highest spatial resolution for defining essential language areas, while SEEG has a complementary role for sulcal and deep network, especially in a paediatric setting or in lesional cases with complex topography, as suggested by Cossu et al [81]. “

“From a surgical perspective, converging data from ECS, SEEG, and ECoG should guide re-sections beyond predetermined surgical margins, highlighting functional boundaries derived from customised, language-appropriate tasks [84,85] . Functional atlases for language mapping can be further improved by combining intraoperative and intracranial (sEEG or subdural) test results”

“The reasons why paediatric results frequently differ from adult data can be attributed to cortex development during the epileptic process, which can reflect on a neuroplastic reorganisation of fronto-temporal networks[88]. To prevent underestimating or overestimating eloquent regions in younger patients, age-stratified interpretation and stimulation threshold adjustment could be es-sential [89]”

“Cross-linguistic research demonstrates that English-based task paradigms might not transfer to other linguistic systems, as tonal and logographic languages use partially different cortical circuits, such as increased activation of laryngeal motor and occipitotemporal regions [91,92]. Reducing cultural bias and increasing the accuracy of functional localisation are two benefits of adapting mapping tasks to native language structure.”

“One significant step to resolve this heterogeneity is the creation of a standardised language test tailored for short periods of brain stimulation. For instance, Ohlerth et al. showed that both object and action naming tasks can be completed within nTMS/DES temporal constraints (≥50 items per task with ≥80% naming agreement) by validating a comprehensive battery (VAN-POP) in 80 healthy adults across three languages. Before being widely used, these protocols must be validated in clinical populations with epilepsy and correlated with intraoperative and postoperative findings [65].”

Reviewer Comment: "States that standardised testing modalities are needed. Should propose actionable steps: 1. Adopt core task battery (visual/auditory naming, reading, repetition). 2. Standardise stimulation parameters. 3. Integrate multimodal validation (ECS+fMRI+DTI). Avoid unsupported general statements."

Response: We thank the reviewer for raising this important point. To address it, we change the conclusion by integrating the concept of multimodal validation approaches: “We present a systematic review of the specific language task selection and their association with different cortical areas, discussing key factors in mapping, including testing modality, definition of language error, identification of critical language area and considerations regarding different pop-ulations. Despite the application of advanced neurophysiological techniques in language mapping, an individualised approach remains a challenge due to variability in the localisation of language sites, which is more pronounced in the epilepsy population and the lack of a standardised protocol for assessing the constellation of lingual elements that are essential for speech production. By integrating evidence-based task selection and a multimodal approach, the accuracy of language mapping can be optimised along with our understanding of language network organisation. A unified battery of expressive and receptive tasks, combining visual and auditory naming, reading or counting, and repetition, while always considering the patient’s multilingual background and age, should be used. Moreover, multimodal assessment combining invasive and non-invasive tools could increase accuracy and prevent possible mistakes inherent to specific techniques. Overall, refining and standardising the language task implementation process could lead to improved outcomes, ultimately minimising resection-related language impairment. Further multicentre studies using comparable language testing protocols are required to provide insight into the organisation of language networks and provide accurate mapping and better post-surgical outcomes.

Reviewer 3 Report

Comments and Suggestions for Authors

  1. The paper aims to present a systematic review and analyze the current techniques and practices in language mapping with electrical stimulation, focusing on the selection of specific linguistic tasks for different cortical areas and evaluating current evidence supporting their validity in correlation with specific brain areas. The abstract is a bit confused between the motivation and contributions of the proposed review paper.
  2. How did you specify the criteria for including and excluding studies (for example type of language task used during stimulation)?
  3. Separate the related works from the introduction section and summarize your contributions.
  4. What is the specific gap in the literature that this research aims to fill?
  5. Reduce the table 2 caption.
  6. In the result and discussion sections, you have to visually present some charts, curves to validate your survey.
  7. Did the review focus on long-term language outcomes following epilepsy surgery guided by ECS mapping.
  8. Did the review incorporate research on multilingual individuals, and how were the mapping procedures adapted to evaluate multiple languages?

Comments on the Quality of English Language

 The English could be improved to more clearly express the research.

Author Response

Thank you very much for your comments. They have been invaluable in substantially improving the quality, rigour, and clarity of our work. We provide a detailed point-by-point response to each comment, indicating the specific changes made and their locations in the revised manuscript.

Reviewer Comment: "The paper aims to present a systematic review and analyse the current techniques and practices in language mapping with electrical stimulation, focusing on the selection of specific linguistic tasks for different cortical areas and evaluating current evidence supporting their validity in correlation with specific brain areas. The abstract is a bit confused between the motivation and contributions of the proposed review paper."

Response: We thank the reviewer for this helpful observation.  We have revised the abstract with a clearer structure, outlining the rationale, objectives, and main findings more coherently.

“Background: Language mapping is a critical component of epilepsy surgery, as post-operative language deficits can significantly impact patients' quality of life. Electrical stimulation mapping has emerged as a valuable tool for identifying eloquent areas of the brain and minimizing post-surgical language deficits. However, recent studies have shown that language deficits can occur despite language mapping, potentially due to variability in stimulation techniques and language task selec-tion. The validity of specific linguistic tasks for mapping different cortical regions remain inadequately characterized. Objective: To systematically evaluate the validity of linguistic tasks used during elec-trical cortical stimulation (ECS) for language mapping in epilepsy surgery, analyse task-specific re-sponses across cortical regions, and assess current evidence supporting optimal task selection for different brain areas. Methods: Following PRISMA [2020] guidelines, a systematic literature search was conducted in PubMed and Scopus covering articles published from January 2013 to November 2025. Studies on language testing with electrical cortical stimulation in epilepsy surgery cases were screened. Two reviewers independently screened 956 articles, with 45 meeting the inclusion criteria. Data extraction included language tasks, stimulation modalities (ECS, SEEG, ECoG, DECS), cortical regions and language error types. Results: Heterogeneity in language testing techniques across various centres was identified. Visual naming deficits were primarily associated with stimulation of the posterior and basal temporal regions, fusiform gyrus, and parahippocampal gyrus. Auditory naming elicited impairments in the posterior superior and middle temporal gyri, angular gyrus, and fusiform gyrus. Spontaneous speech errors varied, with phonemic dysphasic errors linked to the inferior frontal and supramarginal gyri, and semantic errors arising from superior temporal and perisylvian parietal regions. Conclusions: Task-specific language mapping reveals distinct cortical specializations, with systematic patterns emerging across studies. However, marked variability in testing protocols and inadequate standardization limit reproducibility and cross-center comparisons. Overall, refining and standardizing the language task implementation process could lead to im-proved outcomes, ultimately minimising resection-related language impairment. Future research should validate task-region associations through prospective multicenter studies with long-term outcome assessment.”

Reviewer Comment: "How did you specify the criteria for including and excluding studies (for example type of language task used during stimulation)?"

Response: Thank you for this comment. We added a comprehensive section (2.2. Eligibility Criteria) in the method that explicitly defines inclusion and exclusion criteria using PICO framework. We also added a section of this paragraph to clarify inclusion and exclusion criteria (2.4 Study Selection and Data Extraction):

Reviewer Comment: "Separate the related works from the introduction section and summarise your contributions."

Response: We thank the reviewer for this constructive suggestion. In response, we have revised the Introduction to more clearly distinguish the discussion of previous literature from the rationale of the present study. We reorganised the paragraphs so that prior studies and existing reviews are now presented in a more clearly delineated part of the Introduction. Furthermore, we have added a concise paragraph at the end of the Introduction explicitly outlining the aims and key contributions of our systematic review:

This review aims to systematically evaluate the validity of linguistic tasks used during electrical cortical stimulation (ECS) for language mapping in epilepsy surgery, analyse task-specific responses across cortical regions, and assess current evidence supporting optimal task selection for different brain areas.” 

Reviewer Comment:"What is the specific gap in the literature that this research aims to fill?"

Response: Thank you for this important question. In our opinion, this systematic review fills important gaps in the current literature by providing the first comprehensive and up-to-date synthesis of language task validity across cortical regions, specifically in epilepsy surgery. It integrates both traditional electrical cortical stimulation (ECS) and modern stereo-EEG (SEEG) data from the past decade. Data from 45 studies involving 1,196 participants can help us to identify systematic patterns in task-specific cortical responses and propose potential standardisation recommendations for future multicenter studies. As a summary, our systematic review addresses several critical gaps in the current literature on language mapping during epilepsy surgery. Although previous reviews have examined language mapping, our review is specifically focused on the validity and appropriateness of specific linguistic tasks across different cortical regions in epilepsy surgery during the past decade. We added this paragraph in the introduction section to emphasise on the specific literature gap: “Several studies have investigated cognitive tests in neurosurgery. In a comprehensive review, Hamberger et al. (2007) established the foundations which have guided clinical practice for epilepsy surgery for nearly two decades [18]. However, the field has developed significantly since 2007, especially with the global widespread application of SEEG techniques [8,9,19]. In 2013, De Witte et al. conducted a systematic review of intraoperative language testing [20], but their primary focus was on brain tumour surgery. However, when comparing with epilepsy surgery, the challenges are fundamentally different because of different patterns of cortical reorganisation [21]. Ruis conducted a comprehensive review of intraoperative cognitive tests during awake craniotomy for epilepsy and glioma, though the selection of language tests was not evaluated [22]. Lu et al. explored mapping strategies for counting and picture-naming practices, but primarily restricted to these two tasks [23]. More recently, Reecher et al. reviewed language mapping of electrical stimulation language in paediatric epilepsy [9]. However, there is no recent review exploring the use of SEEG in adult and paediatric epilepsy surgery during the last decade”

Reviewer Comment: "Reduce the table 2 caption."

Response: Thank you for your comment. We understand that the caption looks long, but the table has a lot of abbreviations, and we feel that the explanation should be accessible easily to the readers. We have moved the abbreviations in the following line and reduced the font size so it is clear that the caption is just 2 lines.

Reviewer Comment: "In the result and discussion sections, you have to visually present some charts, curves to validate your survey."

Response: Thank you for your comment. Our manuscript includes visual presentations to validate our systematic review:

  • Figure 1: PRISMA flow chart (956 articles screened → 45 included)
  • Figure 2: Anatomical diagram of language-related brain regions with abbreviations
  • Figure 3: Language mapping diagram showing task-region associations
  • Table 1: Language paradigm descriptions and error definitions
  • Table 2: Complete summary of all 45 studies with stimulation modalities, tasks used, and cortical response sites

These figures and tables systematically present our study selection process, anatomical framework, and synthesised findings. If graphs are required, we would need some more time to prepare them, given the amount of detailed information already included in the tables and figures. A possibility would be the attached figure, but it does not give any extra information for the text. If requested, it can be included as supplementary material.

Reviewer Comment: "Did the review focus on long-term language outcomes following epilepsy surgery guided by ECS mapping."

Response: Thank you for this question. We appreciate the importance of this clinical question but long-term language outcomes were not the primary focus of this systematic review. This topic could be the subject of another systematic review, and we would be  happy to perform it if the journal is interested.

In section 2.2 we mention: “Outcomes (O): Primary outcomes included the identification of cortical regions producing specific language deficits during stimulation and the correlation between linguistic tasks and anatomical regions. Secondary outcomes included stimulation parameters and  error types.”

Reviewer Comment: "Did the review incorporate research on multilingual individuals, and how were the mapping procedures adapted to evaluate multiple languages?"

We have included information on multilingual patients in the discussion only but details on multilingual language testing were not found in the included papers.

Round 2

Reviewer 2 Report

Comments and Suggestions for Authors

All comments have been addressed by the authors, the manuscript can be accepted in this format.

Reviewer 3 Report

Comments and Suggestions for Authors

no comments